# A Japanese Dose of Prasugrel versus a Standard Dose of Clopidogrel in Patients with Acute Myocardial Infarction from the K-ACTIVE Registry

**DOI:** 10.3390/jcm11072016

**Published:** 2022-04-04

**Authors:** Hiroyoshi Mori, Takuya Mizukami, Atsuo Maeda, Kazuki Fukui, Yoshihiro Akashi, Junya Ako, Yuji Ikari, Toshiaki Ebina, Kouichi Tamura, Atsuo Namiki, Ichiro Michishita, Kazuo Kimura, Hiroshi Suzuki

**Affiliations:** 1Department of Cardiology, Showa University Fujigaoka Hospital, Yokohama 227-8501, Japan; mizukamit@med.showa-u.ac.jp (T.M.); atsuo@kt.rim.or.jp (A.M.); hrsuzuki@med.showa-u.ac.jp (H.S.); 2Kanagawa Cardiovascular and Respiratory Center, Department of Cardiology, Yokohama 236-0051, Japan; fukui@kanagawa-junko.jp; 3Department of Cardiology, St. Marianna University School of Medicine, Kawasaki 216-8511, Japan; yoakashi-circ@umin.ac.jp; 4Department of Cardiology, Kitasato University School of Medicine, Sagamihiara 252-0375, Japan; jako@kitasato-u.ac.jp; 5Department of Cardiology, Tokai University School of Medicine, Isehara 259-1193, Japan; ikari@is.icc.u-tokai.ac.jp; 6Department of Laboratory Medicine, Yokohama City University Medical Center, Yokohama 232-0024, Japan; tebina@yokohama-cu.ac.jp; 7Department of Cardiology, Yokohama City University Graduate School of Medicine, Yokohama 236-0004, Japan; tamukou@med.yokohama-cu.ac.jp; 8Department of Cardiology, Kanto Rosai Hospital, Kawasaki 211-8510, Japan; namikiatsuo@kantoh.johas.go.jp; 9Department of Cardiology, Yokohama Sakae Kyosai Hospital, Yokohama 247-8581, Japan; i-michishita@yokohamasakae.jp; 10Department of Cardiology, Yokohama City University Medical Center, Yokohama 232-0024, Japan; c_kimura@yokohama-cu.ac.jp

**Keywords:** aspirin, clopidogrel, prasugrel, P2Y12 inhibitor, bleeding

## Abstract

Background: Dual antiplatelet therapy (DAPT) with aspirin plus P2Y12 inhibitor is used as a standard therapy for patients with acute myocardial infarction (AMI) treated with drug-eluting stents (DESs). In Japan, clopidogrel was the major P2Y12 inhibitor used for a decade until the new P2Y12 inhibitor, prasugrel, was introduced. Based on clinical studies considering Japanese features, the set dose for prasugrel was reduced to 20 mg as a loading dose (LD) and 3.75 mg as a maintenance dose (MD); these values are 60 and 10 mg, respectively, globally. Despite this dose discrepancy, little real-world clinical data regarding its efficacy and safety exist. Methods: From the K-ACTIVE registry, based on the DAPT regimen, patients were divided into a prasugrel group and a clopidogrel group. The ischemic event was a composite of cardiovascular death, non-fatal MI, and non-fatal stroke. The bleeding event was type 3 or 5 bleeding based on the Bleeding Academic Research Consortium (BARC) criteria. Results: Substantially more patients were prescribed prasugrel (*n* = 2786) than clopidogrel (*n* = 890). Clopidogrel tended to be selected over prasugrel in older patients with numerous comorbidities. Before adjustments were made, the cumulative incidence of ischemic events at 1 year was significantly greater in the clopidogrel group than in the prasugrel group (*p* = 0.007), while the cumulative incidence of bleeding events at 1 year was comparable between the groups (*p* = 0.131). After adjustments were made for the age, sex, body weight, creatine level, type of AMI, history of MI, approach site, oral anticoagulation therapy, presence of multivessel disease, Killip classification, and presence of intra-aortic balloon pumping, both ischemic and bleeding events became comparable between the groups. Conclusion: A Japanese dose of prasugrel was commonly used in AMI patients in the real-world database. Both the prasugrel and clopidogrel groups showed comparable rates of 1 year ischemic and bleeding events.

## 1. Introduction

Dual antiplatelet therapy (DAPT) with aspirin and a P2Y12 inhibitor is essential for contemporary percutaneous coronary intervention (PCI) with drug-eluting stents (DESs) in patients with acute myocardial infarction (AMI). Clopidogrel had been widely used as the P2Y12 inhibitor of choice in DAPT since 2006 in Japan. However, a considerable proportion of Japanese patients are reported to be CYP2C19 poor metabolizers (PMs), who can only attain a low concentration of the active metabolite of clopidogrel [1,2].

Prasugrel is a newer P2Y12 inhibitor with a more consistent, rapid, and pronounced inhibition of platelet activity than clopidogrel [3,4,5]. In an initial study from the Trial to Assess Improvement in Therapeutic Outcomes by Optimizing Platelet Inhibition with Prasugrel-Thrombolysis in Myocardial Infarction (TRITON-TIMI38) in patients with acute coronary syndrome (ACS) undergoing PCI, which included a very low proportion of East Asian patients (<1%), prasugrel at a standard dose (loading dose (LD)/maintenance dose (MD): 60/10 mg) showed significantly fewer ischemic events but a higher incidence of bleeding than clopidogrel (LD/MD: 300/75 mg) [6]. Because East Asians are known to have a higher bleeding risk than Western populations, a reduced dose of prasugrel (LD/MD: 20/3.75 mg), compared with the standard dose of clopidogrel (LD/MD: 300/75 mg) in the prasugrel group compared with clopidogrel group for Japanese patients with ACS undergoing PCI (PRASFIT-ACS) showed efficacy and safety [7,8]. Accordingly, a reduced dose of prasugrel was approved in 2014 in Japan, and the Japanese Circulation Society (JCS) guideline recommends a reduced dose of prasugrel (LD/MD: 20/3.75 mg) and standard dose of clopidogrel (LD/MD: 300/75 mg) as class I for both ACS and chronic coronary syndrome (CCS) [9]. However, despite this unique dose setting of prasugrel, little real-world clinical data regarding ischemic and bleeding events in Japanese AMI patients have been collected.

Therefore, we tried to assess the efficacy and safety between prasugrel and clopidogrel using the K-ACTIVE (Kanagawa-Acute Cardiovascular Registry) registry.

## 2. Materials and Methods

### 2.1. Study Subjects

The K-ACTIVE is an observational multicenter registry of AMI that enrolled patients from 52 PCI-capable hospitals in Kanagawa Prefecture, Japan, beginning in October 2015, including large and small, urban and rural, and educational and non-educational hospitals. This registry was approved by the local institutional review board and was registered in the University Hospital Medical Information Network (UMIN) in October 2015 (UMIN000019156). AMI was diagnosed as a ST-elevation myocardial infarction (STEMI) or non-STEMI (NSTEMI) based on the Third Universal Definition of Myocardial Infarction Consensus Document [10]. All consecutive AMI patients who presented to hospitals within 24 h of the onset of symptoms were registered. Each attending hospital was required to submit data to an online database on consecutive patients. A follow-up study of patients was performed based on the medical information available at each study site.

### 2.2. Study Endpoint

Patients treated between October 2015 and December 2019 were included in this study. Based on the initial DAPT regimen, patients were divided into a prasugrel group (prasugrel and aspirin) and a clopidogrel group (clopidogrel and aspirin). The selection and duration of medication, including the DAPT, was left to the attending cardiologist based on the JCS guideline [11]. As patients were included before the focused update of the JCS guideline, the duration of the DAPT was likely to be 1 year for most patients [9]. Oral anticoagulation therapy included both warfarin and direct oral anticoagulation therapy. Atrial fibrillation included paroxysmal, persistent, and continuous types. The efficacy endpoint was a composite of cardiovascular death, non-fatal MI, and non-fatal stroke including both ischemic and hemorrhagic. The safety endpoint was type 3 or 5 bleeding based on the Bleeding Academic Research Consortium (BARC) criteria. Secondary endpoints included a composite of ischemic events (cardiovascular death, non-fatal MI, and non-fatal stroke) and bleeding events (BARC type 3 or 5 bleeding).

### 2.3. Statistical Analysis

Continuous variables were expressed as the mean ± standard deviation or median value (25th–75th percentile), as appropriate. The normality of data was tested with the Anderson–Darling test. Categorical variables were expressed as percentages. Continuous variables were compared using a *t*-test or Wilcoxon test. Categorical variables were analyzed by a Fisher’s exact test or the chi-squared test, as appropriate. The age, sex, Killip classification, creatine, use of oral anticoagulation therapy (OAC), body weight, trans-radial approach, type of AMI, previous MI, use of intra-aortic balloon pumping (IABP), and presence of multivessel disease were included in the adjusted model as confounders. Propensity scores for all patients were estimated using multivariable logistic regression models with the above-mentioned confounders. A propensity analysis was conducted using the inverse probability of treatment weights (IPTW) [12]. The cumulative incidence of efficacy endpoint, safety endpoint, and composited ischemic and bleeding events were expressed by a Kaplan–Meier curve without and with adjustment using IPTW. A subgroup analysis was also performed. The JMP 15 (SAS Institute, Cary, NC, USA) or R (version 3.6.1, R Foundation for Statistical Computing, Vienna, Austria) software programs were used to perform the statistical analyses. *p*-values of <0.05 were considered statistically significant.

## 3. Results

### 3.1. Study Population

Between October 2015 and December 2019, a total of 7583 patients were registered in the K-ACTIVE registry. After excluding 3179 patients with missing data regarding antiplatelet therapy and 728 patients without dual antiplatelet therapy, a total of 3676 patients who had received prasugrel (*n* = 2786) and clopidogrel (*n* = 890) were included in the study population. 

### 3.2. Patient Characteristics

Table 1 shows the patient characteristics in each group. The clopidogrel group was older and had more comorbidities, including hypertension; diabetes; dyslipidemia; hemodialysis; and a history of MI, atrial fibrillation, and OAC therapy, than the prasugrel group. The prevalence of male gender and smoking was lower in the clopidogrel group than in the prasugrel group. Among the laboratory data, significant differences were observed in the low-density lipoprotein (LDL) cholesterol, serum creatinine, and albumin levels; the high-density lipoprotein (HDL) cholesterol and HbA1c values did not differ markedly between the groups. The height and body weight values were lower in the clopidogrel group than in the prasugrel group.

### 3.3. AMI Characteristics

Table 2 shows the characteristics of AMI. The prevalence of STEMI and peak creatine kinase levels were lower in the clopidogrel group than in the prasugrel group. There was no significant difference in the culprit vessel, presence of multi-vessel disease, approach site, use of thrombolysis, extracorporeal membrane oxygenation, or out-of-hospital cardiac arrest between the groups. Coronary artery bypass graft and intra-aorta balloon pumping were selected more frequently in the clopidogrel group, while PCI was selected more frequently in the prasugrel group.

### 3.4. Clinical Outcome

Table 3 shows the in-hospital mortality and unadjusted ischemic and bleeding events. Most of the events had a greater prevalence in the clopidogrel group than in the prasugrel group. The cumulative incidence rate of ischemic events, BARC type 3 or 5 bleeding, and composite events, which were unadjusted and adjusted by IPWT, is shown in Figure 1A–C. Ischemic events and composite events were significantly more frequent in the clopidogrel group than in the prasugrel group before adjustment (*p* = 0.007, *p* = 0.002, respectively), while bleeding events were comparable between the groups (*p* = 0.131). All differences became non-significant after adjustment by IPWT. The results of the subgroup analyses are shown in Figure 2A–C. Significant interactions were observed in the radial approach and hemodialysis for composite events.

## 4. Discussion

Regarding the main findings of this study, a substantial number of patients with AMI were treated with prasugrel in a Japanese real-world registry. Prasugrel was used largely in younger, male STEMI patients with fewer comorbidities than clopidogrel-treated patients. Ischemic and bleeding events were observed to have a similar incidence in both groups, with a numerically greater tendency seen in the clopidogrel group.

Globally, clopidogrel is the most frequently used P2Y12 inhibitor in both ACS and CCS, accounting for about 50% to 80% of cases of P2Y12 inhibitor use worldwide [13,14,15,16,17]. However, our data showed that clopidogrel was used only in 24% of patients, while prasugrel was used in 76% of patients in the Japanese ACS registry. This trend was similarly observed in other Japanese registries [18,19,20,21,22]. According to a study by Akita et al. that investigated 62,737 Japanese ACS patients, 68.1% of patients received prasugrel, while 31.9% received clopidogrel [18]. The dose of prasugrel was basically reduced (LD/MD: 20/3.75 mg) in contrast to the standard dose of clopidogrel (LD/MD: 300/75 mg) in these Japanese real-world practice settings [18,19,20,21,22,23]. The findings of such clinical studies comparing a Japanese dose of prasugrel and a standard dose of clopidogrel in CAD patients are inconsistent among Japanese registry studies [18,19,20,21,22,23]. Some studies have reported that bleeding events are more frequent among patients that have received a Japanese dose of prasugrel, while others have reported that bleeding events are less frequent among patients that have received a Japanese dose of prasugrel [19,20,21,22,23]. The relatively low 1-year cardiac mortality rates of our study as compared to the JAMIR data (1.8%, 3.8%, respectively) may be due to a difference in the AMI condition, as the proportions of patients with Killip grade 2 or greater were different (17.2%, 23.9%, respectively). In terms of the efficacy, these two P2Y12 inhibitors seem to be equivalent [18,19,20,21,22,23]. Our study does not seem to show greatly different results from those of these previous studies. Globally, however, the standard dose of prasugrel is likely to be more efficient than a standard dose of clopidogrel at the cost of safety, as reported in the TRITON-TIMI38 [6]. One of the largest network meta-analyses involving 52,816 patients from 12 randomized trials showed that prasugrel reduced the risk of MI (hazard ratio (HR) 0.81, 95% confidence interval (CI) (0.67–0.98)) and stent thrombosis (HR 0.50, 95% CI (0.38–0.64)), but increased the major bleeding risk (HR 1.26, 95% CI (1.01–1.56)) [24].

The East Asian paradox is a well-known phenomenon wherein East Asian patients have a similar or even lower rate of ischemic events than white patients, despite having a higher level of platelet reactivity during DAPT [7]. Thus, a Japanese dose of prasugrel may be reasonable, as shown in the present and previous studies [8,19]. Ohya et al. reported a further reduced maintenance dose of prasugrel (2.5 mg) for patients with a low body weight (≤50 kg), elderly age (≥75 years old), or renal insufficiency (eGFR ≤ 30 mL/min/1.73 m^2^) [25]. The rate of out-of-hospital definite or probable stent thrombosis was 0% in patients receiving prasugrel at 2.5 mg/day (*n* = 284) and 3.75 mg/day (*n* = 487), while the cumulative 1-year incidence of out-of-hospital major bleeding was not significantly different for either of the groups [25]. This strategy seems reasonable [25]. However, the question of whether a single dose or single strategy fits all Japanese patients remains, as about 65% of East Asian individuals carry a CYP2C19 loss-of-function allele, whereas only 30% of white individuals are carriers.

Tailor-made prescriptions have been attempted in prasugrel treatment. Stent thrombosis is reportedly due in part to a CYP polymorphism underuse of prasugrel [26,27]. For patients with the CYP2C19 loss-of-function (LoF) genotype or intermediate/poor metabolizers, a Japanese dose of prasugrel (LD/MD: 20/3.75 mg) or further reduced dose of prasugrel (LD/MD: 20/2.5 mg) might not be sufficient. A recent international meta-analysis assessed the risk of major adverse cardiovascular events (MACEs) following CYP2C19 LoF genotype-guided prasugrel/ticagrelor versus clopidogrel therapy for ACS patients undergoing PCI (*n* = 16132) [28]. Patients treated with prasugrel or ticagrelor showed a significantly reduced risk of MACEs (risk ratio 0.58; 95% CI 0.45–0.76; *p* < 0.0001) compared with those treated with clopidogrel, despite both groups carrying CYP2C19 LoF alleles [28]. Notably, no significant differences in the risk of MACE were found for the patients carrying CYP2C19 non-LoF alleles (risk ratio 0.91; 95% CI 0.81–1.02; *p* = 0.11). Bleeding events were not significantly different between the groups carrying CYP2C19 LoF alleles (Risk ratio 1.06; 95% CI 0.88–1.28; *p* = 0.55) [28]. The VerifyNow-P2Y12^®^ rapid analyzer, which is a rapid assay that tests platelet activity over 3 min and uses of a combination of ADP and prostaglandin E1 (PGE1) to directly measure the effect of P2Y12 inhibitor on the P2Y12 receptor, is now widely available [29]. Monitoring platelet inhibition helped researchers to decide whether or not to use a reduced dose of prasugrel in the initial Japanese Phase II trial [8,30]. A VerifyNow-P2Y12 value of >208 reaction units (PRU) is generally defined as a high on-treatment platelet reactivity (HPR) and has been shown to be related to stent thrombosis and MI, while a VerifyNow-P2Y12 value of <85 PRU is considered to indicate low on-treatment platelet reactivity [31]. These kinds of precision medicines may be ideal, although they are associated with financial issues [32].

Several limitations associated with the present study warrant mention. First, nearly half of the patients in the K-ACTIVE registry were not included in the current analysis due to a lack of information regarding antiplatelet therapy. Second, our registry lacked information regarding the dose and duration of antiplatelet drugs and P2Y12 inhibitor switching after discharge, which influences both ischemic and bleeding events. Because our study population was gathered from 2015 to 2019, which is before the announcement of the focused update of the JCS guideline, it is highly possible that the duration for DAPT was 1 year in most subjects [9]. Similarly, the prasugrel dose was likely to be 3.75 mg in most of the patients, as the further reduced dose of prasugrel (2.5 mg) was only published in 2018 [25]. Third, this was an observational study, and residual or unmeasured confounding factors are likely to persist. For instance, the baseline characteristics differed considerably between the prasugrel and clopidogrel groups. Ischemic and bleeding events may potentially be related to selective prescribing. Although we performed an IPTW analysis to adjust for potential confounders, this method may not be sufficient to abolish this limitation. Fourth, bleeding and ischemic events might be underreported in registries, but this would have been similar for both groups, and severe and ischemic bleeding events are less likely to be missed. Fifth, information regarding the type of stents (drug-eluting stents or bare-metal stents) which can influence the duration of DAPT was not recorded. Finally, the present study was conducted in 52 institutions in Kanagawa, Japan, so the generalization of our finding to other parts of Japan is unreasonable.

## 5. Conclusions

A Japanese dose of prasugrel was frequently used in AMI patients from the real-world database of the K-ACTIVE registry in Kanagawa, Japan. Both the prasugrel and clopidogrel groups showed comparable rates of 1-year ischemic and bleeding events. Further studies are needed to establish optimized antiplatelet therapy for Japanese AMI patients. 

## Figures and Tables

**Figure 1 jcm-11-02016-f001:**
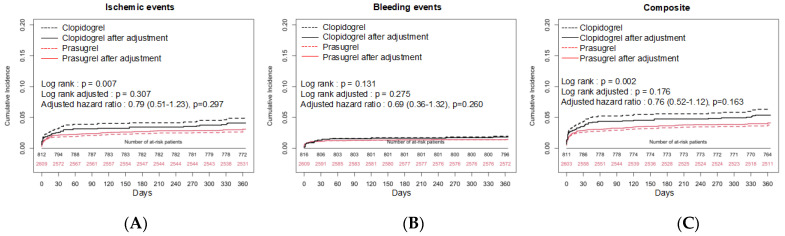
Kaplan–Meier survival curves showing the efficacy endpoint (ischemic events, (**A**)), safety endpoint (bleeding events, (**B**)), and composite endpoint (composite of ischemic and bleeding events, (**C**)) before and after adjustment using the inverse propensity of treatment weights.

**Figure 2 jcm-11-02016-f002:**
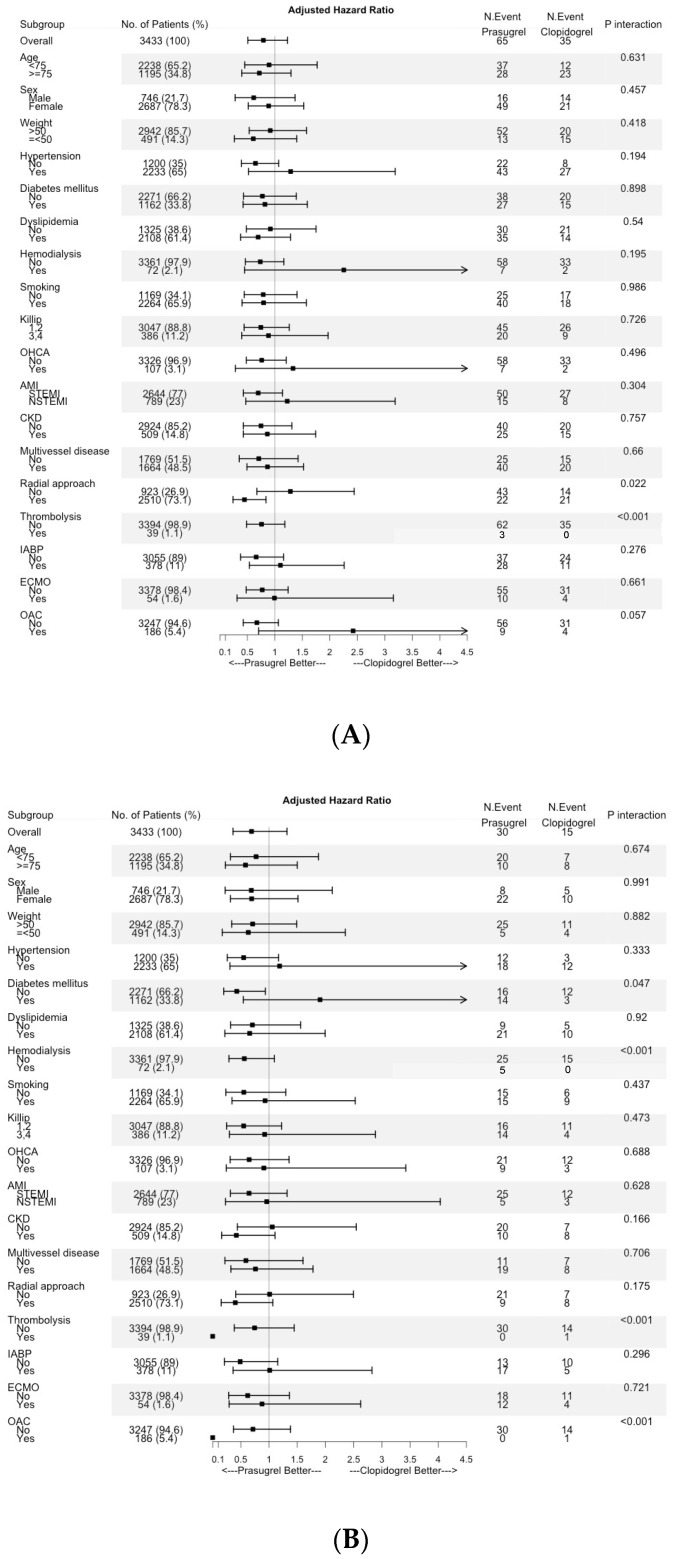
Results of a subgroup analysis of the efficacy endpoint (ischemic events, (**A**)), safety endpoint (bleeding events, (**B**)), and composite endpoint (composite of ischemic and bleeding events, (**C**)).

**Table 1 jcm-11-02016-t001:** Patient characteristics.

	Prasugrel Group (*n* = 2786)	Clopidogrel Group (*n* = 890)	*p*-Value
Age, years	67 ± 16	71 ± 13	<0.01
Male, *n* (%)	2220 (79.7%)	654 (73.5%)	<0.01
Hypertension, *n* (%)	1779 (63.9%)	618 (69.4%)	<0.01
Diabetes, *n* (%)	912 (32.7%)	334 (37.5%)	<0.01
Dyslipidemia, *n* (%)	1052 (37.8%)	363 (40.8%)	0.11
Smoking, *n* (%)	1878 (67.4%)	531 (59.7%)	<0.01
Hemodialysis, *n* (%)	52 (1.9%)	29 (3.3%)	0.02
Previous MI, *n* (%)	225 (8.1%)	126 (14.2%)	<0.01
Atrial fibrillation, *n* (%)	165 (5.9%)	88 (9.9%)	<0.01
Previous hospital visit, *n* (%)	1940 (69.6%)	656 (73.7%)	0.02
Oral anticoagulation therapy, *n* (%)	112 (4.0%)	80 (9.0%)	<0.01
Creatine, mg/dL	0.86 (0.72–1.03)	0.91 (0.76–1.10)	<0.01
LDL, mg/dL	124 (100–151)	114 (90–43)	<0.01
HDL, mg/dL	47 (40–57)	48 (50–58)	0.14
A1c, %	5.9 (5.6–6.6)	6.0 (5.6–6.7)	0.38
Alb, g/dL	4.1 (3.7–4.4)	3.9 (3.6–4.3)	<0.01
Height, cm	165 (158–170)	163 (155–169)	<0.01
Body weight, Kg	65 (56–74)	62 (53–71)	<0.01

Data are expressed as the mean ± standard mediation or median (interquartile) or number (%). MI = myocardial infarction, LDL = low-density lipoprotein cholesterol, HDL = high-density lipoprotein cholesterol, Alb = albumin.

**Table 2 jcm-11-02016-t002:** AMI characteristics.

	Prasugrel Group (*n* = 2786)	Clopidogrel Group (*n* = 890)	*p*-Value
Systolic blood pressure	143 (123–164)	138 (119–162)	<0.01
Heart rate	78 (65–91)	79 (66–92)	0.2
Type of AMI			<0.01
STEMI	2201 (79.0%)	612 (68.8%)	
NSTEMI	585 (21.0%)	278 (31.2%)	
Peak creatine kinase	1503 (601–3224)	1141 (411–2687)	<0.01
Culprit			0.29
LMT	277 (9.9%)	107 (12.0%)	
LAD	1428 (51.3%)	437 (49.1%)	
LCX	151 (5.4%)	51 (5.7%)	
RCA	928 (33.3%)	293 (32.9%)	
Multi-vessel disease	1318 (48.9%)	409 (48.1%)	0.69
Approach			0.41
Radial	1955 (72.4%)	631 (74.3%)	
Femoral	720 (26.7%)	208 (24.5%)	
Brachial	26 (1.0%)	10 (1.2%)	
Percutaneous coronary intervention	2772 (99.5%)	861 (96.7%)	<0.01
Thrombolysis	35 (1.3%)	7 (0.8%)	0.36
CABG	22 (0.8%)	17 (1.9%)	<0.01
IABP	292 (10.5%)	116 (13.1%)	0.04
ECMO	45 (1.7%)	16 (1.8%)	0.76
OHCA	87 (3.1%)	35 (3.9%)	0.24
Killip classification			<0.01
1	2344 (84.1%)	697 (78.3%)	
2	136 (4.9%)	74 (8.3%)	
3	131 (4.7%)	55 (6.2%)	
4	175 (6.3%)	64 (7.2%)	

Data are expressed as median (interquartile) or number (%). AMI = acute myocardial infarction, LM = left main, LAD = left anterior descending artery, RCA = right coronary artery, LCX = left circumflex artery, TIMI = thrombolysis in myocardial infarction, PCI = percutaneous coronary intervention, CK = creatine kinase, IABP = intra-aortic balloon pumping, ECMO = extracorporeal membrane oxygenation, CABG = coronary artery bypass grafting.

**Table 3 jcm-11-02016-t003:** Clinical outcomes.

	Prasugrel Group (*n* = 2786)	Clopidogrel Group (*n* = 890)	*p*-Value
In-hospital mortality	33 (1.2%)	15 (1.7%)	0.24
Ischemic events at 1 year	69 (2.5%)	40 (4.5%)	<0.01
Cardiac death	42 (1.5%)	26 (3.0%)	<0.01
Myocardial infarction	15 (0.5%)	2 (0.2%)	0.39
Stroke	12 (0.4%)	12 (1.4%)	<0.01
Bleeding events at 1 year	24 (0.9%)	14 (1.6%)	0.08

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
