# Peer review of "A Japanese Dose of Prasugrel versus a Standard Dose of Clopidogrel in Patients with Acute Myocardial Infarction from the K-ACTIVE Registry"

_jcm, 2022, doi:10.3390/jcm11072016_

Round 1
Reviewer 1 Report
Thanks for giving me the opportunity to review this manuscript. The authors investigated the efficacy and safety of the Japanese dose of prasugrel compared to clopidogrel. The results suggest that ischemic and bleeding events were comparable in both the Japanese dose of prasugrel and clopidogrel. The manuscript is interesting and well presented, however I have some concerns that needs to be addressed.
Major limitations
#. What is the novel finding of this study compared to previous studies (Ref 17-22)?
#. Although it is a study regarding antiplatelet drug, about half of the enrolled subjects only had information about the antiplatelet drug selection during the hospitalization period, and there were no data on the duration and dosage of antiplatelet drug administration after discharge, and whether there were any antiplatelet drug changes. Therefore, it is difficult to trust the analysis results.
Minor limitations
2.2. Study endpoint
#. There is no information on whether all patients in the Prasugrel group received the same 20mg/3.75mg dose.
#. As this is a study related to the therapeutic effect of antiplatelet agents, stent thrombosis and revascularization should be included in ischemic events. Even if you don't include it as a primary endpoint, you need to show data about it.
#. In this study, it was assumed that the treatment was based on the Japanese guideline. According to the Japanese guideline, anti-coagulation + clopidogrel is the standard when anti-coagulation is used in PCI patients. There are 4% of patients. Because these patients appear to have taken anticoagulants before, it is necessary to determine whether OACs have been discontinued and the change to DAPT with prasugrel is warranted. Finally, Patients with AF or taking oral anticoagulants may affect the end points, so it is desirable to exclude them from the study.
#. I wonder if cardiac death does not include in-hospital deaths. It does not appear to be included numerically, but if so, it may be necessary to discuss why it was excluded when calculating ischemic event and composite event outcomes.
2.3. Statistical Analysis
#. Statistical analysis need to be re-described to be more specific.
#. What was the rationale for choosing Confounder?. It seems that clinically important variables, such as hypertension, diabetes, smoking, hemodialysis, showing significant differences between groups were not included.
#. It would be good to show the baseline characteristics (Table 1 & Table 2) between the two groups after propensity matching.
#. It is necessary to mention which statistical method was used for the subgroup analysis
- Results
#. In what statistical method was Figure 2 analyzed?
#. Please present the overall Hazard ratio (95% confidence interval) and p-value of Figure 2A (ischemic event).
#. The overall of Figure 2C (composite event) seems to be statistically significant, but the adjusted log-rank p-value of Figure 1C, which analyzes the same causal relationship, is 0.138 meaningless. I think you should explain what caused the difference.
- Discussion
#. As the author mentioned references 17 to 22, there are a lot of Japanese data on similar subjects, so it is necessary to discuss the differences between this study and other studies.
#. It would be rather irrelevant to include so much about PRU in the discussion. I think it should be described in relation to this study.
#. Finally, English editing is required.
Reviewer 2 Report
The authors investigate the effect of a "Japanese dose" of prasugrel versus a standard dose of clopidogrel. This is a retrospective study, the authors correct for baseline differences between groups. It would also be important to see a subanalysis in patients with STEMI, and one excluding patients treated without PCI.
Round 2
Reviewer 1 Report
I appreciate your responses to the reviewers' comments, but some issues remain.
There were insufficient data of the duration and dosage of prasugrel after discharge, large portion of missing data and unreliable data regarding outcome data. Therefore, it is difficult to trust the analysis results.
In this study, the 1-year cardiac mortality rates for 42/2786 (prasugrel) and 26/890 (clopidogrel) were 68/3676=1.8%. In the JAMIR data, the average CV death rate in median f/u 12month data is 3.8%. In other ACS registry data in Japan, the in-hospital CV death rate was 3.6% (clopidogrel) and 3.0% (prasugrel). Although the authors said that this study lacked f/u because all AMIs were registered, but had the advantage that it was not intentionally registered like other studies, this study actually showed a low mortality rate. The method of obtaining death data should be described, and it seems necessary to discuss the lower mortality rate compared to other studies in Japan.
Author Response
I appreciate your responses to the reviewers' comments, but some issues remain.
Q1. There were insufficient data of the duration and dosage of prasugrel after discharge, large portion of missing data and unreliable data regarding outcome data. Therefore, it is difficult to trust the analysis results.
A1. We agree that insufficient data limited our study a lot. In comparing two groups. insufficient data would be similar for both groups. We have added to limitation. Bleeding and ischemic event might be underreported in registries, but this would have been similar for both groups, and severe and ischemic bleeding events are less likely to be missed.
Q2. In this study, the 1-year cardiac mortality rates for 42/2786 (prasugrel) and 26/890 (clopidogrel) were 68/3676=1.8%. In the JAMIR data, the average CV death rate in median f/u 12month data is 3.8%. In other ACS registry data in Japan, the in-hospital CV death rate was 3.6% (clopidogrel) and 3.0% (prasugrel). Although the authors said that this study lacked f/u because all AMIs were registered, but had the advantage that it was not intentionally registered like other studies, this study actually showed a low mortality rate. The method of obtaining death data should be described, and it seems necessary to discuss the lower mortality rate compared to other studies in Japan.
A2. We have added a sentence to discuss lower mortality rate and the method of obtaining follow up data. Relatively low 1-year cardiac mortality rates of our study as compared to JAMIR data (1.8%, 3.6%, respectively) may be due to difference in AMI condition as proportion of Killip grade 2 or greater were different (17.2%, 23.9%, respectively). A follow-up study of patients was performed based on the medical information available at each study site.
